# Studying the Pathophysiology of Parkinson’s Disease Using Zebrafish

**DOI:** 10.3390/biomedicines8070197

**Published:** 2020-07-07

**Authors:** Lisa M. Barnhill, Hiromi Murata, Jeff M. Bronstein

**Affiliations:** David Geffen School of Medicine at UCLA, Department of Neurology and Molecular Toxicology Program, 710 Westwood Plaza, Los Angeles, CA 90095, USA; lbarnhill@ucla.edu (L.M.B.); muratahrm@gmail.com (H.M.)

**Keywords:** Parkinson’s disease, zebrafish, toxins, dopaminergic neuron

## Abstract

Parkinson’s disease is a common neurodegenerative disorder leading to severe disability. The clinical features reflect progressive neuronal loss, especially involving the dopaminergic system. The causes of Parkinson’s disease are slowly being uncovered and include both genetic and environmental insults. Zebrafish have been a valuable tool in modeling various aspects of human disease. Here, we review studies utilizing zebrafish to investigate both genetic and toxin causes of Parkinson’s disease. They have provided important insights into disease mechanisms and will be of great value in the search for disease-modifying therapies.

## 1. Introduction

Parkinson’s disease (PD) is a neurodegenerative disorder leading to severe disability, affecting millions of people worldwide [1]. The clinical features of PD reflect the progressive loss of neurons throughout the nervous system. Dopaminergic neuron loss leads to many of the classic motor symptoms of PD, such as resting tremors and slowness of movement, but the pathology is much more widespread and results in many non-motor symptoms as well. Current treatments can improve the symptoms of PD but there are no therapies that alter the progression of the disease. It is essential to understand the pathophysiology of PD before developing disease-modifying therapies and zebrafish (ZF) offer unique qualities to add to this understanding.

## 2. What Is Known About the Pathophysiology of PD

The hallmark pathological finding in PD brains is the presence of intracytoplasmic inclusions called Lewy Bodies (LBs) [2]. Although the vast majority of cases of PD are sporadic, it was the discovery of a mutation in the α-synuclein (α-syn) gene in a family cluster that led to the identification of α-syn as the major component of LBs, both in this family but also in sporadic cases [3,4]. It is widely accepted that the aggregation and propagation of misfolded α-syn underlies the pathogenesis of PD [5,6]. When proteins misfold, they can form aggregates that lead to cell death. Several recently discovered lines of evidence support the self-propagation and spread of α-syn, leading to a predictable progression of PD [2,7]. α-Syn pathology appears to start in the gut and olfactory bulb, then spreading to the substantia nigra (SN) and other brain regions [2]. This process of templating was first described for prion disease but now has been proposed to underlie most neurodegenerative disorders [5,8].

Understanding the causes of protein misfolding and aggregation is essential for understanding the pathogenesis of the disease and are summarized in Figure 1. Increased expression of α-syn by gene duplication or promoter variations is sufficient to cause or increase the risk of developing PD [9,10,11,12,13]. α-Syn levels can also increase through disruption of its degradation. Both the ubiquitin proteasome system and autophagy degrade α-syn [14,15,16,17,18,19] and dysfunction of both of these processes have been implicated in PD. For example, mutations in genes that code for proteasome proteins (e.g., UCHL-1 and Parkin) and proteins involved in autophagy (e.g., GBA and LRRK2) markedly increase the risk of PD [20,21].

Mitochondrial dysfunction has also been implicated as a pathway leading to α-syn aggregation and PD [22,23,24,25]. The causes of mitochondrial dysfunction are diverse and can include both genetic (e.g., PINK1 and Parkin) and environmental (e.g., rotenone and TCE) insults [26,27,28,29]. There is also evidence that α-syn aggregates themselves lead to mitochondrial dysfunction, further propagating a prion-like cascade [30,31,32,33,34]. When mitochondria are damaged, they can be repaired or replaced. Mitophagy is the process by which mitochondria are targeted and degraded via macroautophagy, so dysfunction in autophagy may also lead to mitochondrial dysfunction.

Neuroinflammation (i.e., microglial and astrocyte activation) also appear to be a factor in the pathogenesis of PD [35,36,37]. Microglia are the resident immune cells of the CNS and play key roles in tissue repair and cellular homeostasis [38]. Inflammatory changes may cause injury, protect against it, or simply reflect neuronal injury [36,38,39,40]. Recent studies support a contributing role of inflammation to neuronal damage [37,40]. It is clear that inflammatory changes (microglial and astrocyte activation) are found in PD brains [40], and genetic alterations in several immune function-related genes (e.g., DJ-1, leucine-rich repeat protein kinase-2 (LRRK2), and HLA-DR) can alter the risk of developing PD [41,42].

## 3. Zebrafish as a Model Organism to Study PD

PD does not naturally occur in animals and since it can take decades to develop in humans, no model can accurately incorporate all aspects of the disease. With that said, different animal models can be utilized to study specific aspects of PD. For example, dopamine neurons can be selectively killed in rodents, resulting in some of the motor features of PD. These are excellent models for testing medicines that can relieve the symptoms of PD but may not prevent its pathogenesis. Other models focus on investigating the underlying molecular mechanisms that lead to PD, such as α-syn aggregation and dopaminergic cell loss. Measuring the perturbation of pathways leading to a disease has become a well-accepted approach in studying chronic disease (i.e., adverse outcome pathways (AOP) [43]). This is where ZF offer particular advantages over many other models [44]. Firstly, the transparent nature of the embryos and larvae allows for the use of non-invasive imaging techniques to study neuronal integrity, proteostasis, mitochondrial functions, and microglial activity in genetically modified fish using florescent reporters. Compared to rodents, ZF are also prolific external breeders. This allows researchers to easily genetically modify fertilized eggs without injury to the parent, minimize variation, and maximize experimental replicates. Furthermore, behavioral assays in ZF larvae can be performed in an automated and high-throughput manner, which can act as a powerful screening tool although the range of behaviors that can be measured is relatively limited [44].

Both genetic and toxin-induced ZF models have been used to study PD. A number of factors need to be considered when reviewing these studies. Most investigators utilize embryos and larvae to take advantage of their transparent nature but we must be cognizant that we are studying a degenerative disease in a developing organism [45]. Toxins can easily be added to the water but since the embryos are rapidly developing, the timing of exposure is very important. For example, some toxins affect notochord development during the first 24 h post fertilization (hpf), which would impair all subsequent behavioral assays. Embryos normally hatch approx. 3 days post fertilization (dpf), and the chorion can act as a barrier to toxins and other treatments added to the water. Some investigators choose to dechorionate the embryos prior to exposure, while others leave them intact [46]. Other considerations include the fact that the ZF blood brain barrier forms after 3 dpf, and the fish sexually differentiate at approx. 21 to 23 dpf [46,47]. When performing and reviewing ZF studies, it is important to consider factors such as these to maximize the benefit of using this model organism.

## 4. Genetic Models Used to Study PD

### 4.1. Synuclein

The first mutation found to cause a rare form of autosomal dominant (AD) PD was in the gene coding for α-syn [3]. This led to the finding that α-syn is the major component of LBs, both in the brains of this family but also in brains of sporadic cases [4]. Increased expression of α-syn by gene duplication is sufficient to cause PD [9,10,11,12] and the REP1 263 allele in the α-syn promoter, which confers a higher level of expression [48], is associated with an increased risk of developing PD [13] and faster progression in those that have it [49]. ZF express three syn genes: sncb, sncg1, and sncg2 (encoding β-, γ1-, and γ2-syn, respectively). They do not express an α-syn orthologue [50] although there is evidence that γ1-synuclein (γ1-syn) has a similar function as α-syn. γ1-Syn is highly expressed in the brain and is developmentally regulated in a similar manner as α-syn [51]. Knockdown of γ1-syn and ZF β-syn leads to hypokinesia and reduced dopamine levels, and expression of human α-syn can reverse these abnormalities [52].

We have transiently expressed γ1-syn in neurons using the *HuC* promoter and found it also forms thioflavin T binding to fibrils like α-syn, with similar kinetics [53]. Overexpression of γ1-syn in ZF neurons leads to the formation of aggregates in vivo and is neurotoxic in a similar manner as α-syn [53,54]. This model is limited by the inherent variability of transient expression and that the *HuC* promoter does not express well in dopaminergic neurons. We have made a stable transgenic line of fish using a Gal4/UAS expression system under the ZF HuC promoter (HuC:Gal4 x UAS:α-syn), but there is a much subtler phenotype (unpublished data). Despite these limitations, overexpression of α-syn in ZF neurons has proven to be useful in testing potential drugs that lower the neurotoxicity of α-syn aggregation [53,54].

### 4.2. LRRK2

Mutations in the leucine-rich repeat kinase 2 (LRRK2) gene are the most common cause of autosomal dominant PD and account for 1% of sporadic cases and 4% of familial cases [55]. The G2019S mutation is the most common, especially in Ashkenazi Jewish people or North African Berbers, but there are several other pathogenic variants. The penetrance is generally considered low (approx. 25%) but varies by the population studied [56]. Importantly, patients with LRRK2 mutations present with similar symptoms as idiopathic cases and most contain α-syn containing LBs [56].

LRRK2 is a protein with several domains, including both a serine/threonine kinase and a GTPase domain [21]. It is likely that all or most of the pathogenic mutations in the LRRK2 gene results in a toxic gain-of-function increase in kinase activity and kinase inhibition is considered a promising therapeutic drug target for both LRRK2-induced and idiopathic PD [21].

ZF contain a homologue of the human LRRK2 (hLRRK2) gene, and the protein contains all the functional domains of the human protein. The kinase domain is particularly conserved in ZF with a 71% homology [57]. Despite the fact that gain-of-function increase in kinase activity is considered the most likely mechanism whereby LRRK2 mutations lead to PD, ZF biologists have focused on knocking down the gene in ZF. Morpholino (MO) knockdown of the LRRK2 in ZF (zLRRK2) results in embryonic lethality with severe morphological and neuronal defects, including loss of tyrosine hydroxylase (TH)-positive neurons [57]. The effects of targeted deletion of the Trp-Asp-40 (WD) domain of zLRRK2 using MO is less clear. Sheng et al. [57] reported that this knockdown resulted in a Parkinson’s phenotype including loss of TH+ neurons and locomotive dysfunction, while Ren et al. could not reproduce these findings despite using the same reagents [58]. 

More recently, Prabhudesai et al. knocked down a full length zLRRK2 in a dose-dependent manner and reported generalized morphological defects, loss of *HuC* and dopaminergic neurons, as well as increased levels of β-synuclein, PARK13, and SOD1. They also reported aggregation of β-synuclein; however, this is less clear since no staining for β-pleated sheets was performed and only low power images were shown [59]. These data confirm previous reports of the importance of LRRK2 in general development but also suggests it plays a role in proteostasis. No one yet has reported on the effects of increased kinase activity in ZF, which have a much more direct relevance to the etiology of PD in patients carrying a mutation in the LRRK2 gene.

The function of LRRK2 was further investigated by creating a mutation in the WD40 domain in zLRRK2 [60]. This mutation led to the fish being hyperactive and exhibiting a weakened antibacterial response. Transcriptome analysis revealed that this mutation in zLRRK2 altered the expression of the genes involved in infectious, immunological, and neurological diseases [60]. These studies add insight into why people with LRRK2 variants suffer from a higher incidence of Crohn’s disease and leprosy.

### 4.3. GBA

Gaucher’s disease (GD) is a relatively common autosomal recessive lysosomal storage disease caused by mutations in the glucocerebrosidase 1 (GBA) gene. GBA1 is a lysosomal enzyme required for the breakdown of glucosylceramide to ceramide and glucose and patients exhibit accumulation of glucocerebroside in the spleen, liver, and bone marrow. Clinically, GD presents in a very heterogenous manner. Patients can be asymptomatic or present with severe neurological decline with systemic problems. Several pathogenic mutations have been reported and heterozygote carriers are at an increased risk of developing PD [61]. Depending on the mutation, the risk of developing PD can increase from 2 to 19 fold [20]. GBA mutations result in loss of enzymatic activity of glucocerebrosidase, leading to lysosomal dysfunction that is believed to be responsible for the increased risk of PD. Patients with GBA mutations and PD have classic LBs, suggesting that the increased risk in these patients is due to a decreased degradation of α-syn through autophagy [20].

This concept that GBA mutations lead to an increased risk of PD by reducing α-syn degradation has been directly challenged by Keatinge et al. using their ZF GBA model [62]. They deleted 23 bp of the GBA orthologue that resulted in many of the characteristics of PD. Early on, they found sphingolipid accumulation, up regulation of miR-155 (a regulator of inflammation), and microglial activation. At 8 weeks post-fertilization (wpf), they reported decreased motor activity, dopaminergic cell loss, mitochondrial dysfunction, altered autophagy markers, and ubiquitin-positive intra-neuronal inclusions. Interestingly, all of these PD-like pathological changes occurred in the absence of α-syn and the ZF β- and γ1-synuclein levels were decreased. Furthermore, ablation of miR-155 expression did not alter microglial activation or the disease pathology [63]. Other features of GD were also apparent.

Uemura et al. knocked out GBA in medaka and reported some but not all of the findings of Keatinge et al. [64]. They reported infiltration of Gaucher-like cells into the brain, progressive aminergic neuronal loss, and microgliosis. Interestingly, they observed accumulation of α-syn in autophagosomes but knocking out α-syn in GBA-deficient medaka did not improve survival or the axonal swellings. Taken together, GBA deficiency results in dysfunction in autophagy as well as the accumulation of sphingolipids and proteins, such as α-syn; however, the α-syn is not necessary for neuronal loss.

### 4.4. Parkin

Mutations in the Parkin gene is the most common cause of autosomal recessive (AR) PD [55]. It is important to note that most PD patients with Parkin mutations do not have an LB pathology, even though they show fairly classic clinical signs of early onset PD [65]. For this reason, the pathological pathways leading to dopaminergic cell loss and clinical PD are likely different than the pathological pathways responsible for the majority of idiopathic or most other forms of genetic PD. The Parkin gene encodes for an E3 ligase that targets many proteins, including soluble α-syn, for degradation by the UPS or lysosomes. Parkin also plays an important role in targeting damaged mitochondria for clearance via mitophagy [65].

The ZF Parkin gene was cloned, and the protein was predicted to be 62% identical to the human Parkin protein and 78% identical in the most important functional regions [66]. MO knockdown of ZF Parkin led to reduced mitochondrial complex 1 activity and a 20% reduction in diencephalic dopaminergic neurons at 3 dpf. The authors reported that the Parkin knockdown ZF were more sensitive to MPP+ but the additional dopaminergic loss appeared to be additive. The fish swam normally at 5 dpf but had abnormal electron-dense material in the T tubules in muscle tissue [66]. 

Fett et al. also studied Parkin in ZF and described increased aggregation of Parkin under conditions of oxidative stress or in the presence of dopamine [67]. They used antisense technology to reduce the Parkin levels by 53%, but saw no morphological or behavioral alterations, and the dopamine neuron counts were normal at 3 dpf. Finally, they reported that overexpression of Parkin using a ubiquitous promoter reduced the number of apoptotic cell death while reduced levels of Parkin increased the death after heat shock determined by acridine orange.

### 4.5. (PTEN)-Induced Putative Kinase 1 (Pink1)

The 2nd most common cause of AR PD is mutations in the Pink1 gene, resulting in young onset disease [65]. LBs have been reported in two cases while they were absent in one report [68]. Pink 1 is a highly conserved 581 aa protein that is widely expressed in the CNS and associates with mitochondrial membranes. It is believed to play an important role, along with Parkin, to regulate mitochondrial quality, and the loss of function of Pink1 leads to disease [69].

MO knockdown of Pink1 has led to variable results in ZF. An early study reported severe developmental malformations, a reduction in dopaminergic neurons, and mitochondrial dysfunction [70]. Interestingly, Pink1 knockdown resulted in increased in GSK3β activity and GSK3β inhibitors partially rescued the malformations. In a study by Xi et al., MO knockdown of Pink1 did not lead to loss of dopamine neurons but altered patterning of their projections and altered locomotion [71]. Sallinen et al. reported that Pink1 protein was expressed in dopaminergic neurons and MO knockdown did not result in reduced dopaminergic neurons but did sensitize them to MPTP [72]. 

Flinn et al. took a different approach to study Pink1 [73]. They identified a fish line with mutant Pink1 from a mutagenized library. The knockout fish had a 25% reduction in dopaminergic neurons, which appeared to be relatively specific and not reflective of a more generalized developmental defect. They also observed microglial activation. The mitochondria in these fish had an abnormal morphology and reduced complex I and III activity. Taking a transcriptomic approach, they found that *TigarB* was highly upregulated and knockdown of *TigarB* reversed the mitochondrial defects caused by the loss of Pink1. Taken together, Pink1 appears to be important in maintaining dopaminergic neurons and mitochondrial function. The developmental malformations described by Anichtichik et al. were likely due to off-target effects.

### 4.6. DJ1

Mutations in the gene coding for DJ1 (PARK7) is another cause of AR PD, but LBs have yet to be described in brains of these patients [65]. The DJ1 protein is widely expressed and mostly localizes in the cytosol, as well as in the nucleus and mitochondria. Its function is not completely clear, but it appears to be able to sense oxidative stress and regulate anti-oxidant and anti-apoptotic gene expression [65].

The ZF orthologue of DJ1 is 83% identical to the human protein, and is expressed in the brain, muscle, and gut [74]. MO knockdown of DJ1 did not lead to loss of dopaminergic neurons, but did increase their sensitivity to H_2_O_2_ and proteasome inhibition [75]. Interestingly, overexpression of DJ1 in ZF astroglia led to increased expression of several genes related to oxidative stress in a similar pattern as nrf2 and was protective against MPTP toxicity [76].

## 5. Toxins and the Study of PD

There are a variety of reasons to study toxins in ZF with respect to PD. They can be used to create an anatomical model of PD by killing dopamine neurons. Others use them to study the biological plausibility of an association between exposure and disease. ZF are especially helpful in studying the mechanisms of action of toxins (i.e., adverse outcome pathways) and how they may alter the risk of PD. In particular, two different ZF lines, DAT-EGFP and VMAT2-EGFP, have been very useful in these studies to determine the effects of toxins on dopamine neurons [77,78]. Other studies utilized whole mount in situ hybridization or immunohistochemistry for labeling TH.

### 5.1. Toxins That Kill Dopamine Neurons

MPTP (1-methyl-4-phenyl-1,2,3,6-tetrahydropyridine) is one of the most commonly used toxins to kill dopamine neurons. It was discovered when addicted individuals injected the synthetic opioid MPPP that was contaminated with MPTP, and suddenly became Parkinsonian [79,80]. Its mechanism of toxicity is due to the enzymatic conversion of MPTP by MAO-B to MPP+, which is selectively taken up by the dopamine transporter (DAT) in dopamine neurons. This leads to the inhibition of mitochondrial complex I, which results in free radical formation, reduced ATP synthesis, and, ultimately, death of the neuron. MPTP-treated animals have been very useful as a model to test dopaminergic medications, but the mechanism of cell death is not necessarily related to that in PD [79].

Several groups have reported that MPTP treatment kills the dopamine neurons in ZF. The dose and timing of exposure likely accounts for the variability in reported effects. A single intraperitoneal injection of MPTP in adult ZF resulted in transient decreases in brain dopamine levels and locomotion, but not loss of dopamine neurons [81]. Embryonic and larval ZF are more sensitive to MPTP and exposure for 2–3 days beginning at 24 hpf results in loss of dopamine neurons and reduced locomotion [82,83,84,85]. The mechanism of toxicity of MPTP is the same in ZF as in mammals, since MAO-B and DAT inhibition block toxicity [83,84]. There were some morphological abnormalities when embryos were exposed to higher concentrations of MPTP [85].

6-Hydroxydopamine (6-OHDA) is another toxin used to kill dopamine neurons. It is taken up by DAT and kills neurons by a mechanism involving oxidative stress. Injection of 6-OHDA into adult ZF results in a small reduction in dopamine and locomotion, but no apparent loss of dopamine neurons [81]. Embryonic exposure to 6-OHDA resulted in a small reduction in dopaminergic neurons, but no change in locomotion [85].

### 5.2. Toxins Associated With the Pathogenesis of PD

Exposure to a number of environmental toxins, especially pesticides, have been associated with an increased risk of developing PD and ZF have been a valuable tool in determining if these associations represent causality. 

#### 5.2.1. Rotenone Is a Mitochondrial Complex I Inhibitor and Is Associated With an Increased Risk of PD

Systemic administration of rotenone in rats leads to α-syn accumulation, loss of dopamine neurons, and motor deficits. Systemic administration of rotenone in adult ZF had no effect of dopamine neurons or locomotion [82] but others have reported decreased dopamine, locomotion, and olfaction when put in the water [86,87]. Rotenone exposure to embryos results in a moderate loss of dopamine neurons, decreased locomotion, and occasional cardiac defects. There was no determination of the selectivity of the neuronal loss [85].

#### 5.2.2. Paraquat Is Another Pesticide Associated With an Increased Risk of Developing PD

Paraquat is very similar to MPTP structurally, which is why it was initially studied. It has since been determined that, unlike MPTP, it is not a substrate for DAT or a complex I inhibitor but a redox cycler, and enhances oxidative stress in dopamine neurons [88]. In mammals, exposure to paraquat leads to an approximately 20% decrease in dopaminergic neurons and evidence of oxidative stress [89]. Dopamine neuron loss was greatly enhanced when used in combination with the fungicide maneb [90]. Interestingly, epidemiological studies have shown the risk of PD is also enhanced when exposed to maneb in addition to paraquat [91].

Treating ZF with paraquat has had mixed results. Bretaud et al. found no effect on embryos that were treated from 24 hpf to 5 dpf at concentrations up to 10 mg/L [82]. Nellore and Nandita reported decreased locomotion, dopamine, and serotonin, and evidence of oxidative stress when the embryos were treated with low dose paraquat from 18 to 96 hpf [92]. When ZF were treated with 1 mM paraquat from 3 to 7 dpf, Kalyn found a 16% decrease in dopamine neurons as well as decreased DAT and TH expression, but no change in behavior [85]. In adult ZF, IP injection every 3 days (total of six injections) of paraquat led to decreased locomotion but increased dopamine concentration, no change in TH expression, and decreased DAT expression [93]. When placed in the water for 4 weeks, paraquat had no effect on adult ZF [82].

#### 5.2.3. Ziram Is a Dithiocarbamate Fungicide, and Is an E1 Ligase Inhibitor of the UPS

Exposure to ziram is associated with an increased risk of developing PD [91,94]. The biological plausibility of a causal association was tested using ZF embryos exposed to 50 nM of ziram at 24 hpf, resulting in selective loss of dopaminergic neurons and altered swimming in the dark in a similar manner to dopamine blockage [53]. Interestingly, the dopamine neuron loss was γ1-syn-dependent, since knockdown with MO was protective. Furthermore, CLR01, a drug that breaks apart γ1-synuclein fibrils, was also protective [53].

#### 5.2.4. Benomyl Is Another Fungicide Found to Be Associated With an Increased Risk of Developing PD

Similar to ziram, it also killed dopamine neurons in a selective manner in ZF [95,96]. The mechanism of toxicity was found to be due to inhibition of aldehyde dehydrogenase that detoxifies the dopamine metabolite DOPAL [95,96].

#### 5.2.5. Air Pollution Has Recently Been Found to Be Associated With an Increased Risk of PD and Alzheimer’s Disease, Although the Mechanisms Remain Largely Unknown

Diesel exhaust particle extracts (DEPe), commonly used as a surrogate model of air pollution in health effects studies, was used to determine the biological plausibility and mechanisms of toxicity of this association. ZF embryos treated with DEPe for 24 h (24 to 48 hpf) and analyzed at 5 dpf resulted in loss of dopaminergic as well as non-dopaminergic neurons, and altered behavior [97,98]. Using a transgenic ZF line that measures neuronal autophagic flux [99], it was found that DEPe inhibited flux and that the enhancers of autophagy were protective of neuronal loss.

## 6. Conclusions

Animal models are essential for the study of disease mechanisms, as they allow us to determine the causes and lead us towards the discovery of better treatments. ZF offer several advantages over mammalian models in that that they are inexpensive, transparent, and easily manipulated genetically. Here, we reviewed many of the studies utilizing ZF that investigated genetic and environmental causes of PD. They have provided new insights into the pathogenesis of PD that have been extended into mammalian models. Future ZF studies will likely include high-throughput screens to discover the environmental toxins associated with PD as well as novel therapeutics to treat the disease. 

## Figures and Tables

**Figure 1 biomedicines-08-00197-f001:**
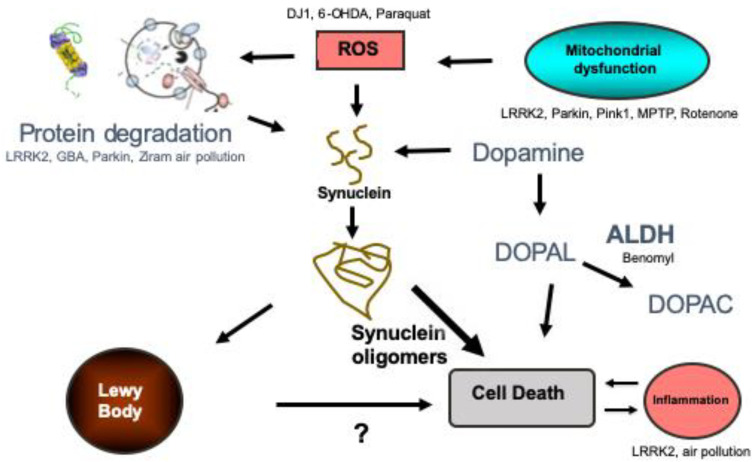
Summary of the proposed pathogenic pathways leading to Parkinson’s disease. The genes and toxins that have been studied in zebrafish are listed in the pathways that they likely influence. Protein degradation refers to autophagy and the ubiquitin proteasome system. ROS refers to reactive oxygen species. ALDH refers to aldehyde dehydrogenase. DOPAL refers to 3,4-Dihydroxyphenylacetaldehyde and DOPAC is 3,4-Dihydroxyphenylacetic acid.

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
