# Peer review of "Studying the Pathophysiology of Parkinson’s Disease Using Zebrafish"

_biomedicines, 2020, doi:10.3390/biomedicines8070197_

Round 1

Reviewer 1 Report

In the review manuscript "Studying the pathophysiology of Parkinson's Disease using zebrafish" Barnhill and colleagues provide an extensive and very well-written review of the literature regarding the use of zebrafish as a model for the study of PD. I find the work sound and very timely, since the use of this model for neurodegenerative diseases is at the same time very promising (for its applications both in the investigation of molecular mechanisms and in the implementation of high-throughput drug screening assays) and somewhat controversial (modelling human neurodegenerative diseases often linked with ageing in the developing larva of a Teleost). The review very well introduces these critical aspects and presents zebrafish for its fundamental advantages in the field. I find this work a very useful contribution in this area of research, providing a comprehensive review of pathogenic mechanisms and related ZF models.

On a general ground I would recommend, if possible, drawing a figure with a general (albeit as simple as the authors deem) scheme of the proteins, intracellular pathogenic mechanisms and genes involved. I think this would provide a summary and a guideline for the reader which would greatly improve the readability of the paper offering a unifying vision of all described mechanisms.

On a technical ground I have only one question regarding LRRK2: the authors state (p. 3, line 31) that pathogenic mutations result om toxic gain-of-function increase of kinase activity. Yet, when describing ZF models (lines 33-40) results are reported of experiments with morpholino knockdown, which would model a loss-of-function but is reported as causative of PD phenotype.

Correction to the text: p.4, line 16 "is believed to responsible", a "be" is missing.

Author Response

Thank you for your helpful comments.  We have addressed them and made the requested changes to the manuscript as detailed below.

Reviewer 1.

“On a general ground I would recommend, if possible, drawing a figure with a general (albeit as simple as the authors deem) scheme of the proteins, intracellular pathogenic mechanisms and genes involved.”

Response:   We agree that a figure summarizing the pathogenic mechanisms would be helpful and we added a cartoon (Figure 1).

“the authors state (p. 3, line 31) that pathogenic mutations result om toxic gain-of-function increase of kinase activity. Yet, when describing ZF models (lines 33-40) results are reported of experiments with morpholino knockdown, which would model a loss-of-function but is reported as causative of PD phenotype.”

Response:  The reviewer makes an excellent point and is clearly a limitation in the presented studies.  To highlight this limitation, we added language on line 132 and 145 to make this point clear.

“Correction to the text: p.4, line 16 "is believed to responsible", a "be" is missing.”

Response:  Corrected.

Reviewer 2 Report

Summary: The authors have summarised the literature surrounding zebrafish models of PD pertaining to the pathophysiology in the form of a review to be submitted to ‘biomedicines’. Overall, the manuscript is a good summary of the pathophysiology of PD and the factors contributing to the development of disease. At current, the manuscript needs further work before consideration for publication in the current format. It would be good to discuss the contribution of infection to the development of PD and PD-like disease. The authors consistently failed to include references throughout the text which led to many unsubstantiated claims within the literature and would not serve its primary role as a review of the published literature and a collection of important references within the field. There are several grammatical and typographical errors in the manuscript which I have highlighted that should be fixed. Moreover, the review would benefit substantially from the inclusion of a summary figure which broadly emphasises the major findings in the field, however this is not a requirements and merely a suggestion to improve the impact of this work.

Minor comments:

Age 1 Line 21: Why focus only on the American individuals? PD affects millions globally, I think a global estimate would be more appropriate and appeal to a greater readership.

Page 2 Lines 3-4: Where is the reference to support this statement?

Page 2 Line 8: Change lead to ‘leads’

Page 2 Lines 12-19: Is there any evidence for the link of infection-induced inflammation and PD? If so, please discuss this here and throughout the review where appropriate.

Page 2 Line 30: You have used embryo and larvae in the same sentence – these are often used interchangeable to reflect the same stage of development. I would suggest choosing one and using throughout.

Page 2 Lines 30-49: You have extensively highlghted many important aspects of the development of the zebrafish embryo yet you have no references to support your statements. Please insert appropriate references.

Page 3 Lines 13-21: Formatting is obscure. Please amend.

Page 3 Lines 23-32: Please insert references where factual statements are made.

Page 4 Lines 5-6: This is a stretch. Suggest revising this statement.

Page 4 Lines 8-18 and 37-44: Insert references

Page 5 Lines 8-12: Insert references

Page 5 Line 20: Change Flinn et all to ‘et al’

Page 5 Lines 30-34: Insert references

Page 6 Lines 2-11: Insert references

Page 6: Line 2: I would strongly suggest changing the term ‘drug addicts’ to a more appropriate term.

Page 6: Subheadings under 5.2 ‘Toxins associated with the pathogenesis of PD’ possess unusual titles and would be more appropriate if titled as the major factor eg ‘Rotenone’ instead of the very long current title.

Page 6 Line 36: There is a mistake written as ‘Structurally, it very similar’ which should be changed.

Page 6 Line 43: There is a mistake written as ‘Bretaud et al found no effect of when’ which should be changed.

Author Response

Thank you for your helpful comments.  We have addressed them and made the requested changes to the manuscript as detailed below.

Reviewer 2.

 “It would be good to discuss the contribution of infection to the development of PD and PD-like disease”.

Response:  This is a review about zebrafish models of Parkinson’s disease (PD).  We did summarize what are generally felt to be the predominant pathogenic processes leading to PD mainly to give context to the zebrafish literature reviewed in our report.  There is little to no evidence that infections play a direct role in the development of PD and there are no zebrafish studies that I am aware of that have studied this.  I am not sure what PD-like diseases are being referred to. There are several disorders that affect the basal ganglia and produce similar symptoms of PD but the pathophysiology of these disorders vary considerably.  Parkinson-plus disorders and secondary Parkinsonisms are outside the scope of this review.

“The authors consistently failed to include references throughout the text which led to many unsubstantiated claims within the literature and would not serve its primary role as a review of the published literature and a collection of important references within the field.”

Response:  As mentioned above, our intent was to provide a brief summary of the pathogenic processes leading to PD mainly to give context to the zebrafish literature reviewed in our report.  We agree that this summary was not well referenced and 21 additional references to substantiate all statements.  With this said, we did not intend this summary to serve as a complete review of the published literature but only to give context to the primary intent of reviewing the ZF literature.

“There are several grammatical and typographical errors in the manuscript which I have highlighted that should be fixed.”

Response:  Thank you for pointing out these errors.  The ones listed below have been corrected as have a few more we found.  If there is a highlighted version of our manuscript with more errors, we would be happy to correct them but we do not have access to this version.

 “Moreover, the review would benefit substantially from the inclusion of a summary figure which broadly emphasises the major findings in the field, however this is not a requirements and merely a suggestion to improve the impact of this work.”

Response:   We agree and a figure has been added.

Minor comments:

“Age 1 Line 21: Why focus only on the American individuals? PD affects millions globally, I think a global estimate would be more appropriate and appeal to a greater readership.”

Response:  This has been changed

“Page 2 Lines 3-4: Where is the reference to support this statement?”

Response:  References 3 and 4 have been added

“Page 2 Line 8: Change lead to ‘leads”

Response:  done

“Page 2 Lines 12-19: Is there any evidence for the link of infection-induced inflammation and PD? If so, please discuss this here and throughout the review where appropriate.”

Response:  Inflammation has been implicated in PD and is briefly summarized but since there has been essentially no ZF studies published on CNS inflammation with respect to PD, we did not go into this in detail because the purpose of the pathological review of PD was to give context to the ZF studies.   There are associations with remote infections such has chronic hepatitis but causality has not been demonstrated and more importantly for this report, has not been studied in ZF.

Page 2 Line 30: You have used embryo and larvae in the same sentence – these are often used interchangeable to reflect the same stage of development. I would suggest choosing one and using throughout.

Response:  Embryos refer to the offspring prior to hatching and larvae refer post-hatching fish so they can not be used interchangeably.  For example, larvae can be used for behavior but embryos cannot.

“Page 2 Lines 30-49: You have extensively highlghted many important aspects of the development of the zebrafish embryo yet you have no references to support your statements. Please insert appropriate references.”

Response:  Since this a review to be part of a special issue on zebrafish, we thought that reviewing the basic biology of zebrafish would be redundant.  We did add another general reference (#44) in response to your comment and the specific models used in the studies reviewed are thoroughly referenced.

“Page 3 Lines 13-21: Formatting is obscure. Please amend.”

Response:  Corrected

“Page 3 Lines 23-32: Please insert references where factual statements are made.”

Response:   References added

“Page 4 Lines 5-6: This is a stretch. Suggest revising this statement.”

Response:  We respectfully disagree and have added a reference

“Page 4 Lines 8-18 and 37-44: Insert references”

Response:   References added

“Page 5 Lines 8-12: Insert references”

Response:   References added

“Page 5 Line 20: Change Flinn et all to ‘et al’”

Response: Changed

“Page 5 Lines 30-34: Insert references”

Response:   There are only 28 lines on page 5 so we are not sure what the reviewer is referring to.  This section is extensively referenced.

“Page 6 Lines 2-11: Insert references”

Response:   This study was referenced but at the end of the paragraph.   We added the reference earlier in the paragraph as to avoid any confusion.

Page 6: Line 2: I would strongly suggest changing the term ‘drug addicts’ to a more appropriate term.

Response:   The term was changed to “addicted individuals” which was taken from the original description of MPTP by Dr. Langston and colleagues (ref #80).

“Page 6: Subheadings under 5.2 ‘Toxins associated with the pathogenesis of PD’ possess unusual titles and would be more appropriate if titled as the major factor eg ‘Rotenone’ instead of the very long current title.”

Response:  We respectfully disagree.  “Toxins associated with the pathogenesis of PD” accurately describes this section and is not a very long title.  We discuss 5 different toxins that have been implicated in the pathogenesis of PD and have been studied in ZF.

“Page 6 Line 36: There is a mistake written as ‘Structurally, it very similar’ which should be changed.”

Response:  This has been changed.

“Page 6 Line 43: There is a mistake written as ‘Bretaud et al found no effect of when’ which should be changed.”

Response:   This has been corrected.